# Investigation on the Enzymatic Profile of Mulberry Alkaloids by Enzymatic Study and Molecular Docking

**DOI:** 10.3390/molecules24091776

**Published:** 2019-05-08

**Authors:** Zhihua Liu, Ying Yang, Wujun Dong, Quan Liu, Renyun Wang, Jianmei Pang, Xuejun Xia, Xiangyang Zhu, Shuainan Liu, Zhufang Shen, Zhiyan Xiao, Yuling Liu

**Affiliations:** 1State Key Laboratory of Bioactive Substance and Function of Natural Medicines, Beijing Key Laboratory of Drug Delivery Technology and Novel Formulation, Institute of Materia Medica, Chinese Academy of Medical Sciences & Peking Union Medical College, Beijing 100050, China; liuzhihua0207@163.com (Z.L.); dwujun@vip.sina.com (W.D.); wry@imm.ac.cn (R.W.); pangjm@tidepharm.com (J.P.); xjxia@imm.ac.cn (X.X.); 2Beijing Key Laboratory of Active Substance Discovery and Drug Ability Evaluation, Institute of Materia Medica, Chinese Academy of Medical Sciences and Peking Union Medical College, Beijing 100050, China; yangying@imm.ac.cn (Y.Y.); xiaoz@imm.ac.cn (Z.X.); 3Pharmacology and Natural Medicine Research Laboratory, Institute of Materia Medica, Chinese Academy of Medical Sciences & Peking Union Medical College, Beijing 100050, China; popliu@imm.ac.cn (Q.L.); liusn@imm.ac.cn (S.L.); shenzhuf@imm.ac.cn (Z.S.); 4Beijing Wehand-Bio Pharmaceutical Company Limited, 30 Tianfu Street, Beijing 102600, China; zhuxiangyang68@163.com

**Keywords:** type 2 diabetes mellitus, mulberry alkaloids, α-glucosidase inhibitors, kinetics analysis, molecular docking

## Abstract

α-glucosidase inhibitors (AGIs) have been an important category of oral antidiabetic drugs being widely exploited for the effective management of type 2 diabetes mellitus. However, the marketed AGIs not only inhibited the disaccharidases, but also exhibited an excessive inhibitory effect on α-amylase, resulting in undesirable gastrointestinal side effects. Compared to these agents, Ramulus Mori alkaloids (SZ-A), was a group of effective alkaloids from natural *Morus alba* L., and showed excellent hypoglycemic effect and fewer side effects in the Phase II/III clinical trials. Thus, this paper aims to investigate the selective inhibitory effect and mechanism of SZ-A and its major active ingredients (1-DNJ, FA and DAB) on different α-glucosidases (α-amylase and disaccharidases) by using a combination of kinetic analysis and molecular docking approaches. From the results, SZ-A displayed a strong inhibitory effect on maltase and sucrase with an IC_50_ of 0.06 μg/mL and 0.03 μg/mL, respectively, which was similar to the positive control of acarbose with an IC_50_ of 0.07 μg/mL and 0.68 μg/mL. With regard to α-amylase, SZ-A exhibited no inhibitory activity at 100 μg/mL, while acarbose showed an obvious inhibitory effect with an IC_50_ of 1.74 μg/mL. The above analysis demonstrated that SZ-A could selectively inhibit disaccharidase to reduce hyperglycemia with a reversible competitive inhibition, which was primarily attributed to the three major active ingredients of SZ-A, especially 1-DNJ molecule. In the light of these findings, molecular docking study was utilized to analyze their inhibition mechanisms at molecular level. It pointed out that acarbose with a four-ring structure could perform desirable interactions with various α-glucosidases, while the three active ingredients of SZ-A, belonging to monocyclic compounds, had a high affinity to the active site of disaccharidases through forming a wide range of hydrogen bonds, whose affinity and consensus score with α-amylase was significantly lower than that of acarbose. Our study illustrates the selective inhibition mechanism of SZ-A on α-glucosidase for the first time, which is of great importance for the treatment of type 2 diabetes mellitus.

## 1. Introduction

Non-insulin-dependent diabetes mellitus (type 2 diabetes, DM) has become a serious health concern characterized by hyperglycemia due to inadequate production of insulin, causing complications such as cardiovascular diseases, retinopathy and nephropathy. Growing evidence suggests that elevated postprandial glucose is closely associated with the occurrence of DM [1,2]. To date, in the categories of oral antidiabetic drugs, α-glucosidase inhibitors (AGIs) are well received among Asian populations, which can significantly delay the intestinal carbohydrate digestion and reduce the postprandial blood glucose levels by inhibiting α-glucosidase (maltase, sucrase, and α-amylase) located in the intestinal brush border [3,4]. However, the excessive inhibition of α-amylase by AGIs will result in gastrointestinal side effects because of an increase of undigested carbohydrate and intestinal fermentation. Currently, the most widely used AGI in clinics is the first product named acarbose, which exhibits excellent hypoglycemic effect. Unfortunately, acarbose can give rise to major adverse effects such as diarrhea, flatulence and vomiting due to excessive inhibition of α-amylase [5,6]. Additionally, according to the α-glucosidase selectivity, the monocyclic compounds have been developed into products such as miglilol (Figure 1, compound 8), which exhibit fewer adverse effects than acarbose, but the adverse gastrointestinal events still exist. Therefore, compared with the currently marketed AGIs, natural AGIs with monocyclic structures from plants sources, which have lower α-amylase inhibitory activity with minor side effects, can be a promising and effective therapy for postprandial hyperglycemia. 

Currently, a number of natural products with superior properties that are capable of inhibiting α-glucosidase in vitro and in vivo have received considerable attention. The extracts of mulberry (*Morus alba* L.) have been reported to have strong inhibitory effect on α-glucosidase, which can effectively reduce postprandial hyperglycemia [7,8,9]. By consulting the related literature, as the most studied constituent in extracts, it was found that 1-DNJ exhibits an effective inhibitory activity on α-glucosidase [10]. However, most of the research is limited to the enzymatic activity assay and determination methods, and few reports are available on the selective inhibition mechanism of 1-DNJ on α-glucosidase compared to acarbose [11,12]. At present, Ramulus Mori alkaloids extracted from a traditional Chinese herb (Mori Ramulus, the sticks from the mulberry plant), abbreviated as SZ–A, are being developed to be a novel hypoglycemic drug. The active components of SZ-A is a group of alkaloids belonging to monocyclic compounds, including 1-deoxynojirimycin (1-DNJ), fagomine (FA), 1,4-dideoxy-1,4-imino-d-arabinitol (DAB), 3-epi-fagomine, 2-o-β-Glc-DAB, 6-o-β-d-Glc-DNJ, and 2-o-β-Glc-DAB (Figure 1, compounds **1** to **7**) [13], the total alkaloid content in SZ-A extract is about 55%, the major active ingredients in SZ-A are 1-DNJ, FA and DAB, which account for more than 80% of the total alkaloids. Moreover, SZ-A has been approved for clinical trials under the investigation of the Institute of Materia Medica, Chinese Academy of Medical Sciences. The results of Phase II/III clinical trials (registered at http://www.chinadrugtrials.org.cn/, numbers CTR20140569 and CTR20140034) showed SZ-A possessed a remarkable effect in reducing glycosylated hemoglobin with fewer side effects such as diarrhea and intestinal inflation compared with the positive control of acarbose. The clinical results suggested that SZ-A displayed a selective inhibitory effect on disaccharidases, which had great potential for clinical applications. Nevertheless, there were no systematic studies on the selective inhibitory effect and molecular binding mechanism of SZ-A and its major active ingredients (1-DNJ, FA and DAB) on disaccharidases and α-amylase. Therefore, it is of great significance to research the selective inhibitory activity and mechanism of SZ-A and its major active ingredients (1-DNJ, FA and DAB) in detail. Accordingly, the enzyme kinetic studies could provide useful information on the selectivity and pattern of enzyme inhibition effect in vitro. Computer-aided molecular modelling methods such as molecular docking have been proposed as a valuable tool for exploring the protein–ligand binding mode, which plays a significant role in unravelling the molecular basis of disease and drug discovery [14,15,16]. Thus, enzyme-kinetics and molecular docking may be conducive to revealing the selective inhibition mechanism on α-glucosidase of AGIs.

In our study, we provide a reliable strategy to understand the selective inhibitory effect and mechanism of SZ-A on disaccharidases and α-amylase. First, taking acarbose (Figure 1, compound **9**) as positive control, we tested the α-glucosidase inhibitory profile of SZ-A and its major active ingredients (1-DNJ, FA and DAB) by enzyme inhibition assay and kinetic studies in vitro. Then, molecular docking was performed to provide valuable insights into the binding properties between the major active ingredients (1-DNJ, FA and DAB) and the α-glucosidase [17,18,19]. There were four kinds of α-glycosidases involved in carbohydrate degradation including pancreatic amylase (HPA), maltase-glucoamylase N-terminal subunit (NtMGAM), maltase-glucoamylase C-terminal subunit (CtMGAM), and sucrase-isomaltase N-terminal subunit (NtSI) [20,21,22], which were used for molecular docking. Meanwhile, the consensus scoring function was utilized to objectively evaluate the docking results of alkaloids. The consensus scoring function, combination of multiple scores, might dramatically reduce the number of false positives by its distinct scoring functions. The consensus number of votes and consensus normalized average values were regarded as the criteria to evaluate the docking results. This paper aims to illustrate the selective inhibitory effect and mechanism of SZ-A on α-glucosidase using a combination of kinetic analysis and molecular docking approaches, which also provide some evidence for the clinical applications of SZ-A and the development of hypoglycemic products.

## 2. Results and Discussion

### 2.1. Inhibition of α-Glucosidase by SZ-A

The inhibitory properties of SZ-A and the isolated active compounds on α-glucosidase were analyzed and IC_50_ values were shown in Table 1. As shown in Table 1, for sucrase and maltase, 1-DNJ exhibited a potent inhibitory effect on sucrase and maltase with IC_50_ values consistent with the published literature [23]. Even though their structure is similar to that of 1-DNJ, the other two compounds (DAB and FA) showed less inhibitory activity. In addition, the IC_50_ of SZ-A on maltase and sucrase was 0.06 μg/mL and 0.03 μg/mL, respectively. The IC_50_ value of acarbose was 0.07 μg/mL and 0.68 μg/mL after conversion according to the molecular weight (mol wt = 645.6). Accordingly, 1-DNJ, as the highest content of SZ-A, showed a notable inhibition activity with an IC_50_ of 0.02 μg/mL and 0.01 μg/mL on maltase and sucrase (mol wt = 163.17), respectively. Apparently, the inhibitory effect of SZ-A is mainly due to the 1-DNJ and is even stronger than that of acarbose. In contrast, the inhibitory effects on α-amylase showed an entirely different pattern. Acarbose strongly inhibited α-amylase, whereas 1-DNJ, FA, DAB, and SZ-A exhibited no obvious inhibitory activity. These results suggest that the inhibitory activity on α-glucosidase of SZ-A derived from Morus is different from that of acarbose and exhibits higher selectivity. In addition, the inhibitory effect of 1-DNJ on sucrase and maltase was 1282 times and 885times more potent than that of amylase, respectively (IC_50_ amylase/IC_50_ disaccharidases), while acarbose was only 2.57 times and 25.87 times more potent than that of amylase. Thus, the lack of amylase inhibition indicates that SZ-A causes less undesirable side-effects and possesses a very high therapeutic value.

### 2.2. Enzyme-Kinetic Analysis

To better understand the inhibition mechanism between these compounds and α-glucosidase, the α-glucosidase kinetics were measured in the presence of different concentrations of inhibitors. As shown in Figure 2 and Figure 3, all curves of the Lineweaver–Burk LB plots of compounds intersected on the *y*-axis, indicating that 1-DNJ, FA, DAB, and acarbose induced competitive inhibition on maltase and sucrase. The kinetic parameters (*Km*, *Ki*) were calculated and shown in Table 2. As the concentration of alkaloids increased, the *Km* value increased, indicating that enzyme catalysis requires more substrates and inhibitory effect is higher. The *Ki* values of 1-DNJ for maltase and sucrase are 1.12 × 10^−6^ M and 1.37 × 10^−8^ M, respectively, which confirms that it tends to be more liable to associate with disaccharidase. The inhibitory activity sequence of different compounds on sucrase and maltase is obtained by comparing the *Ki* values from the LB plots as follows: 1-DNJ > acarbose > FA > DAB, 1-DNJ ≈ acarbose > DAB > FA. These results are consistent with the corresponding IC_50_ values, which shows 1-DNJ exhibits stronger inhibitory activity on disaccharidase. 

### 2.3. Molecular Docking Studies

#### 2.3.1. Catalytic Analysis of α-Glucosidases

The major active ingredients in SZ-A (1-DNJ, FA and DAB) and the positive controls (acarbose and miglitol) were docked into the four α-glycosidases to evaluate the binding modes at the molecular level. Prior to molecular docking, we compared the amino acid sequence and structure of HPA, NtMGAM, CtMGAM, and NtSI. As shown in Figure 4, the sequence similarity of NtMGAM with NtSI and CtMGAM was close to 60% and 30%, respectively, whereas the similarity between HPA and other structures was only about 15%. These results are in agreement with literature reports [24] and show that the sequence and structure of HPA are quite different from those of MGAM and SI, and that sequence homology and structural similarity between MGAM and SI are higher. 

To further evaluate the structural similarities between different α-glucosidases, the SAS of each enzyme was calculated and the results are shown in Table 3. SAS values of MGAM and NtSI were basically the same and significantly higher than that of HPA, which revealed that the binding cavities of MGAM and SI are smaller and located in a more accessible area. Compared with N-terminal domains (NtMGAM and NtSI), C-terminal domains possess a larger binding cavity, which may accommodate the compounds with higher molecular weight. HPA possessed the largest binding cavity and could accommodate more than five sugar rings by forming hydrogen bonds, and the active pocket of HPA was buried deeper than that of MGAM and SI.

#### 2.3.2. Molecular Docking and Consensus Scoring

Four crystal structures were used for docking study, and the details of the docking models are shown in Table 4. The binding pocket was defined with a default sphere radius around the co-crystalized ligand of the crystal structure. Then, the co-crystalized ligand was re-docked into the crystal structure by using the Glide program. The value of Root Mean Square Deviation (RMSD) was calculated to evaluate the reliability of the docking algorithms. The result indicates the reliability of the docking algorithm.

The major active ingredients in SZ-A (1-DNJ, FA and DAB), and the positive control (acarbose and miglitol) were subsequently subjected to molecular docking to verify the binding mode. Then docking compounds with the best binding modes were evaluated by the consensus scoring function. The result is shown in the Appendix A. In our study, by adopting the consensus number of votes and consensus normalized average values provided by the DS, the results of 12 scoring programs were comprehensively evaluated and are shown in Table 5. Additionally, the interactions of these compounds with HPA, NtSI, NtMGAM and CtMGAM were analyzed for identifying binding residues, which are shown in Figure 5, Figure 6, Figure 7 and Figure 8, respectively. 

From Table 5, it can be seen that the positive control of acarbose achieved high consensus scores for both disaccharidases and HPA, indicating that acarbose had strong binding affinity to various kinds of α-glucosidase. Furthermore, the result of docking study and consensus score evaluation agreed with the inhibition assay results described above, which suggested the above results were credible. Additionally, the major active ingredients (1-DNJ, FA and DAB) obtained consensus scores for the disaccharidases (NtMGAM, CtMGAM and NtSI) which were much higher than that of HPA, which indicated that these compounds formed a favorable binding mode with disaccharidases to some extent. Moreover, the docking results of these compounds were in agreement with the inhibition assay research. Thus, these major active ingredients are effective selective inhibitors of disaccharidases and therefore play a major role for SZ-A to reduce blood glucose. 

To be specific, according to the consensus score results of compounds with HPA, acarbose ranked first. In the 3D docking results of the crystal structures, it was observed that acarbose could form a wide range of hydrogen bonding with catalytic amino acid residues such as Gln63, Lys200, Thr163, Trp59, Asp300, His305, Glu240 and Gly306. Acarbose also showed hydrophobic interaction with Leu162 (Figure 5A). In addition, 1-DNJ (compound **1**) formed hydrogen bonds with Glu233, Asp197, His299, and Asp300 at the active pocket (Figure 5B). In addition, FA (compound **2**) could form hydrogen bonds with Arg195, Asp197 and His299 (Figure 5C), and DAB (compound **4**) formed hydrogen bonds with Glu233, Asp300 and Asp197 (Figure 5D). The results suggest that acarbose had the strongest binding affinity with HPA owing to its structural characteristics, which could occupy the binding cavity more effectively than the other compounds.

To contrast the docking results of NtSI, the major active ingredients (1-DNJ, FA and DAB) all obtained decent consensus scores compared with the positive control of acarbose and miglitol. More specifically, 1-DNJ, FA and DAB could perform hydrogen bond interactions with Asp472, Asp571 and Asp355. Compared to the related literature, Asp571 and Asp355 are known to be key residues in the NtSI active site, which was consistent with our study [25]. Detailed docking results of DNJ, FA and DAB are shown in Figure 6A–C, respectively. According to literature reports [26], the hydrolysis of sucrose is mainly catalyzed by the C-terminal domains of sucrase (CtSI), whereas NtSI is responsible for the hydrolysis of isomaltose. Therefore, the enzyme inhibition results described above may be related to the interaction of compounds with CtSI. Because no crystal structure information is currently available for CtSI, we did not perform docking studies of molecules with this subunit, and further research is needed to prove the consistency of the docking simulation to evaluate inhibition of this enzyme.

MGAM is a therapeutic target in the treatment of type 2 DM. According to the docking result of NtMGAM, the positive control of miglitol formed hydrogen bonds with Asp327, Asp542, His600 and Arg526 (Figure 7A), which was consistent with literature [27]. In addition, 1-DNJ obtained the highest score and the binding mode between 1-DNJ and NtMGAM were analyzed. As shown in Figure 7B, 1-DNJ could form hydrogen bonds with Asp443, Asp542, His600, Asp327, and Arg526 in the binding pocket, which was similar to the interactions formed by miglitol. Meanwhile, FA also formed hydrogen bonds with Asp443, Asp327, His600 and Asp542 (Figure 7C). DAB could form the hydrogen bonds with Asp542, Asp327, Asp443 and Arg526 (Figure 7D). Therefore, the above results suggest that these active ingredients could perform favorable interactions with NtMGAM to reduce blood glucose levels.

Because of the larger cavity size of CtMGAM compared to that of NtMGAM, we found that the binding modes of these compounds with the two domains were entirely different. As seen in Figure 8A, 1-DNJ docked into the active site region of CtMGAM by forming hydrogen bonds with residues Arg1510, His1584, Asp1420, Asp1526 and Asp1279. FA could form hydrogen bonds with Arg1510, His1584, Asp1526, Asp1420 and Asp1279 (Figure 8B). DAB formed hydrogen bonds with Trp1355, Asp1420, Asp1526 and Asp1279 (Figure 8C). In addition, acarbose could form hydrogen bonds with Arg1510, His1584, Asp1526, Asp1157, Asp1420 and Asp1279, also showed hydrophobic interactions with residues Tyr1251, Trp1355, and Phe1559, which had a high affinity for CtMGAM (Figure 8D). Scoring results also showed that acarbose ranked highest among all compounds (Table 4).

NtMGAM and CtMGAM may carry out independent catalytic activities on maltose with different hydrolysis rates [28,29]. CtMGAM shows a preference for longer substrates compared with NtMGAM because of the presence of +2 and +3 subsites and a larger cavity. Our docking results revealed that 1-DNJ, FA and DAB displayed favorable binding affinity for NtMGAM, which was similar to that of the positive control of miglitol. In contrast, these compounds showed weaker binding capacity with CtMGAM compared to that of acarbose. These results are in agreement with recombinant enzyme inhibition tests reported in literature [30]. Our study employed maltase (full length MGAM) for the enzymological analyses instead of NtMGAM and CtMGAM, which influence the activity of the maltase cooperatively. On the basis of the above analysis, the active compounds showed a good correlation between scoring results in the binding mode of disaccharidases (maltase and sucrase) and enzyme inhibition activity in the assays, which further highlighted the importance of molecular mechanic-based scoring function in revealing the biological properties of target compounds. In conclusion, by the combination of inhibition kinetics and molecular simulation approaches, SZ-A and its active compounds exhibited the selective inhibition of α-glucosidase to reduce blood glucose.

## 3. Materials and Methods 

### 3.1. Materials

Sucrase and maltase were obtained from the intestinal brush border of Wistar rats [31]. (Institute of Laboratory Animal Science, CAMS & PUMC, Beijing, China, approved number: No. 00001042). α-Amylase, DAB, and miglitol were purchased from Sigma-Aldrich (St. louis, MO, USA), FA was purchased from MedChem Express (Nanjing, China), SZ-A and 1-DNJ were provided by Institute of Materia Medica, Chinese Academy of Medical Sciences (the content of 1-DNJ in SZ-A was 35%), and acarbose was purchased from J&K Scientific (Beijing, China). Each stock solution was prepared with distilled water. All other solvents and chemicals were of analytical purity. 

### 3.2. α-Glucosidase Inhibition Assay

The modified inhibitory screening system [32] was used to determine inhibitory activity on α-glucosidase. Selected starch, sucrose, and maltose were used as α-glucosidase substrates instead of p-nitrophenyl glucopyranoside. Maltase and sucrase were pre-incubated with different concentrations of compound (0.004–400 μmol/L) at 37 °C for 8 min, and the substrate was then added into initiate the reaction. After incubation at 37 °C for 40 min, 150 μL of glucose oxidase solution was added to each mixture and incubated at 37 °C for 40 min. Absorption at 505 nm was measured with a Microplate Reader (Bio-Tek, Winooski, VT, USA). α-glucosidase inhibitory effects in each sample were determined based on the formation of glucose. Inhibitory percentage values were calculated using the formula:
%Inhibition = (B − C)/B × 100,
where B is the formation of glucose under the enzyme without compound and C is the formation of glucose with compound. 

α-amylase was pre-incubated with different compound concentrations (0.016–100 μmol/L) at 37 °C for 15 min, and the reaction was initiated with the addition of 100 µL of 0.04% starch solution. After incubation at 37 °C for 10 min, 0.1 M iodine reagent was added. Absorption at 660 nm was measured to determine the remaining amount of starch in the reaction and thus calculate the inhibitory effect in the different samples.

### 3.3. Enzyme-Kinetic Analysis

The inhibitory activity of the alkaloids 1-DNJ, FA, and DAB towards α-glucosidase was determined, with acarbose as positive control, using various concentrations of inhibitors with different substrates (sucrose and maltose) and substrate concentrations (0.056–0.278 M). Lineweaver–Burk double reciprocal plots (LB plots) were then obtained to calculate the kinetic constants and evaluate inhibitory type, the equation is as follows:
1/V = (Km + [S])/(Vmax[S]) = (Km/Vmax) * (1/[S]) + (1/Vmax),

In this equation, [S] denotes the concentration of substrate, V represents the enzyme reaction velocity and K_m_ is the Michaelis–Menten constant.

### 3.4. Molecular Docking Studies

To further understand the molecular mechanism for the selective inhibition of α-glucosidase by SZ-A, molecular docking was then performed to reveal the binding modes between alkaloids and α-glucosidase. 

#### 3.4.1. Sequence Alignment and Structural Superposition of Proteins

The available co-crystal structures of α-glucosidases with inhibitors were retrieved from the Research Collaboration for Structural Bioinformatics (RCSB) Protein Data Bank (PDB codes: 1HNY, 3LPO, 2QLY, 3TON, 3LPP, 3L4W, and 3TOP). To obtain sequence and structural information of α-glucosidase proteins by comparison, sequence alignment and structure superimposition were performed using the alignment/superposition tool with default parameters in MOE (Chemical Computing Group, Montreal, QC, Canada). 

#### 3.4.2. SAS Calculations

To determine the extent of exposure to solvent of the α-glucosidase structures, the Solvent Accessible Surface (SAS) area in Å^2^ of crystal structures was calculated using the solvent accessibility tool with default parameters in Discovery Studio v4.0 (DS) from Accelrys (San Diego, CA, USA) [33].

#### 3.4.3. Molecular Docking Studies

Docking calculations were carried out using the Glide program (Schrödinger, LLC: New York, NY, USA, 2007). Generally, the crystal structures of HPA (PDB code 3OLD), human NtMGAM (PDB code 3L4W), CtMGAM (PDB code 3TOP), and NtSI (PDB code 3LPP) were obtained from the RCSB Protein Data Bank. The proteins were automatically cleaned up by the preparation tools for some common problems, such as incomplete residues, the lack of hydrogens and the existence of crystallographic water [34]. The active site of each protein for docking was determined based on the position of the co-crystalized ligand within the binding pocket. After being removed from the active pocket, the co-crystalized ligand was re-docked into the crystal structure. The RMSD value was calculated and used to measure the reasonability of the docking algorithm. Then, the maximum number of docking conformations for each ligand was set to 10. Both the standard precision (SP) mode and the extra precision (XP) mode were employed with the default settings. The OPLS-2005 force field was used for the docking protocol [35]. The default scoring functions corresponding to Glide (XP) were used for the docking score calculation. 

#### 3.4.4. Consensus Scoring

In general, the docking score is used to evaluate the binding mode between the compounds and enzymes. Compared with any single scoring procedure, consensus scoring is more robust and accurate with a moderate number of scoring functions [36]. Therefore, we utilized 12 different scoring programs as follows: (1) LigScore1, LigScore2, PLP1, PLP2, Jain, PMF, and PMF04, provided by the DS; (2) Glide score, provided by the Schrödinger suite program; (3) Surflex_score, Dock-like score, Gold-like score, and ChemScore, provided by the Sybyl program. Assessment and ranking were carried out using consensus scoring and normalized averages.

## 4. Conclusions

To further understand the selective inhibition mechanism of SZ-A on α-glucosidase, a combination of inhibition kinetics and molecular docking simulation were conducted for the first time. The principal results showed that: (i) the main active ingredients in SZ-A (1-DNJ, FA and DAB) exhibited strong inhibitory activity on sucrase and maltase in competitive inhibition, among them 1-DNJ displayed the highest activity with IC_50_ values of 0.078 × 10^−6^ M and 0.113 × 10^−6^ M, respectively, even stronger than that of acarbose; (ii) 1-DNJ, FA, DAB and SZ-A exhibited no inhibitory activity on α-amylase compared with acarbose, which provided the evidence for developing an effective agent to reduce the postprandial hyperglycemia with minimal side effects; (iii) 1-DNJ, FA and DAB achieved decent consensus scores and formed favorable interactions with the active site of MGAM and SI, more so than HPA, which revealed that the selective inhibition mechanism of the major active ingredients in SZ-A on α-glucosidase at the molecular level. In addition, the molecular docking analysis validated the inhibition activity of 1-DNJ, FA and DAB on α-glucosidase, which indicated that our results were reliable. The above results indicate the unique enzymatic profile advantages of SZ-A and its major active ingredients. In the near future, there is potential for follow-up study to further investigate the possible synergistic effects and mechanisms of the different compounds in SZ-A on α-glucosidase. In conclusion, this study illustrates the selective inhibitory effect and mechanism of SZ-A on α-glucosidase and contributes in providing a reference for the clinical applications of SZ-A. 

## Figures and Tables

**Figure 1 molecules-24-01776-f001:**
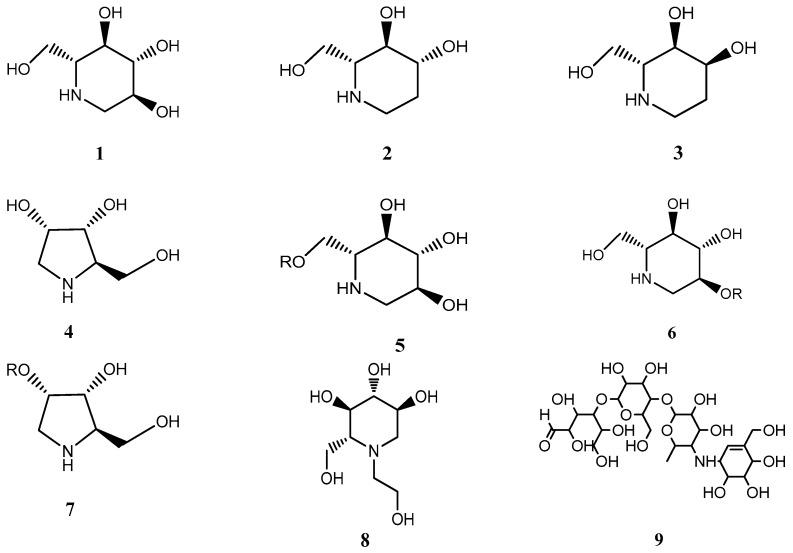
(**1**) Chemical structures of 1-deoxynojirimycin (1-DNJ), (**2**) fagomine (FA), (**3**) 3-epi-fagomine, (**4**) 1,4-dideoxy-1,4-imino-D-arabinitol (DAB), (**5**) 2-o-β-Gal-DNJ, (**6**) 6-o-β-D-Glc-DNJ, (**7**) 2-o-β-Glc-DAB, (**8**) miglitol, (**9**) acarbose.

**Figure 2 molecules-24-01776-f002:**
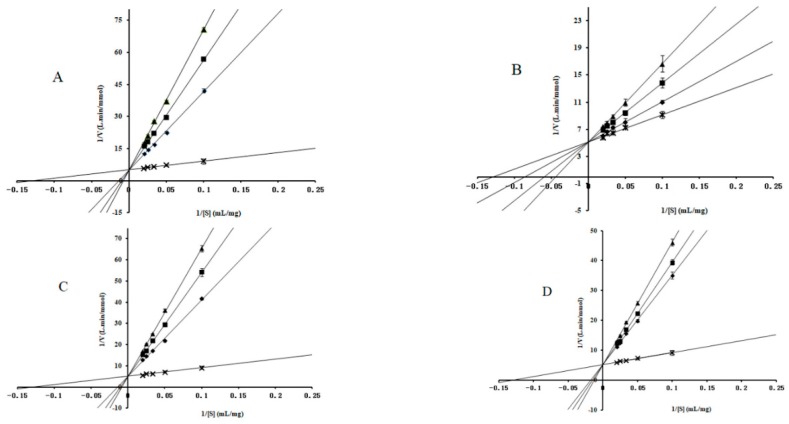
Determination of the inhibition mode of molecules on maltase using Lineweaver–Burk reciprocal plots. (**A**) The concentrations of 1-DNJ were 0 (**×**), 12.5 (◆), 25.0 (■) and 50.0 (▲) × 10^−6^ mol/mL. (**B**) The concentrations of FA were 0 (**×**), 17.0 (◆), 34.0 (■) and 68.0 (▲) × 10^−6^ mol/mL. (**C**) The concentrations of DAB were 0 (**×**), 27.3 (◆), 54.5 (■) and 109.0 (▲) × 10^−6^ mol/mL. (**D**) The concentrations of acarbose were 0 (**×**), 6.3 (◆), 12.5 (■) and 25.0 (▲) × 10^−6^ mol/mL. The concentrations of maltose were 0.028, 0.056, 0.083, 0.111, and 0.138 M. Data are presented as mean ± SD.

**Figure 3 molecules-24-01776-f003:**
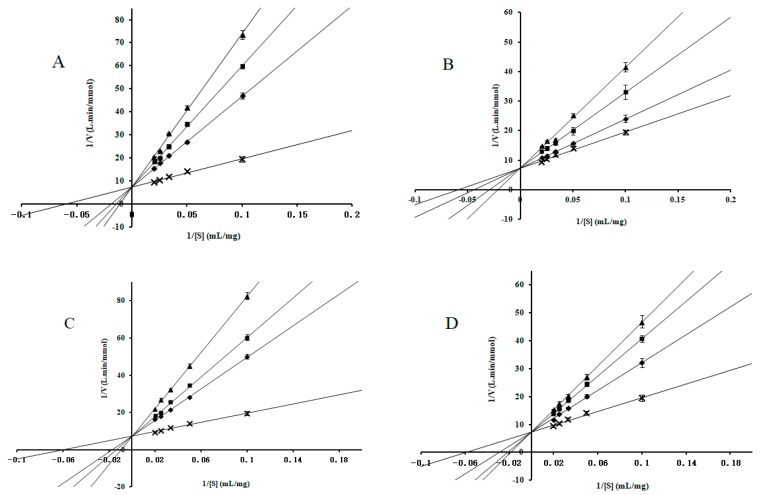
Determination of the inhibition mode of molecules on sucrase using Lineweaver–Burk reciprocal plots. (**A**) The concentrations of 1-DNJ were 0 (**×**), 3.1 (◆), 6.3 (■) and 12.5 (▲) × 10^−8^ mol/mL. (**B**) The concentrations of FA were 0 (**×**), 17.0 (◆), 34.0 (■) and 68.0 (▲) × 10^−6^ mol/mL. (**C**) The concentrations of DAB were 0 (**×**), 10.9 (◆), 21.8 (■) and 43.6 (▲) × 10^−6^ mol/mL. (**D**) The concentrations of acarbose were 0 (**×**), 6.3 (◆), 12.5 (■) and 25.0 (▲) × 10^−8^ mol/mL. The concentrations of sucrose were 0.029, 0.058, 0.088, 0.117, and 0.146 M. Data are presented as mean ± SD.

**Figure 4 molecules-24-01776-f004:**
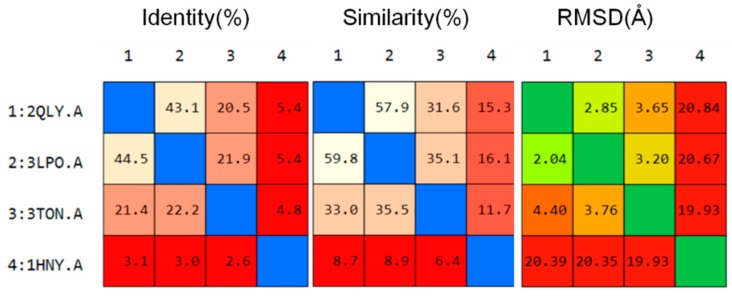
Sequence alignment and structure superposition matrix of maltase-glucoamylase N-terminal subunit (2QLY), sucrase-isomaltase N-terminal subunit (3LPO), maltase-glucoamylase C-terminal subunit (3TON), and pancreatic amylase (1HNY).

**Figure 5 molecules-24-01776-f005:**
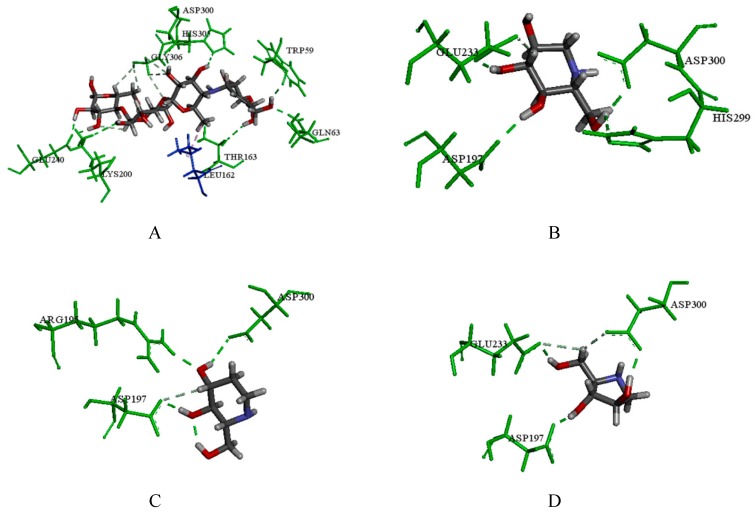
(**A**) The docking result of acarbose with the crystal structure of HPA; (**B**) the docking result of 1-DNJ; (**C**) the docking result of FA; (**D**) the docking result of DAB; the green amino acids represent hydrogen bond interactions; blue amino acids represent hydrophobic interactions.

**Figure 6 molecules-24-01776-f006:**
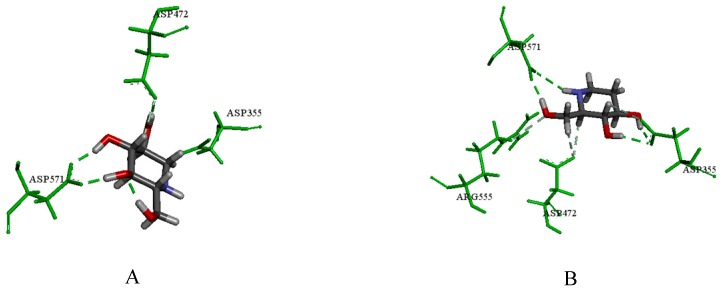
(**A**) The docking result of 1-DNJ with the crystal structure of NtSI; (**B**) the docking result of FA; (**C**) the docking result of DAB; the green amino acids represent hydrogen bond interactions.

**Figure 7 molecules-24-01776-f007:**
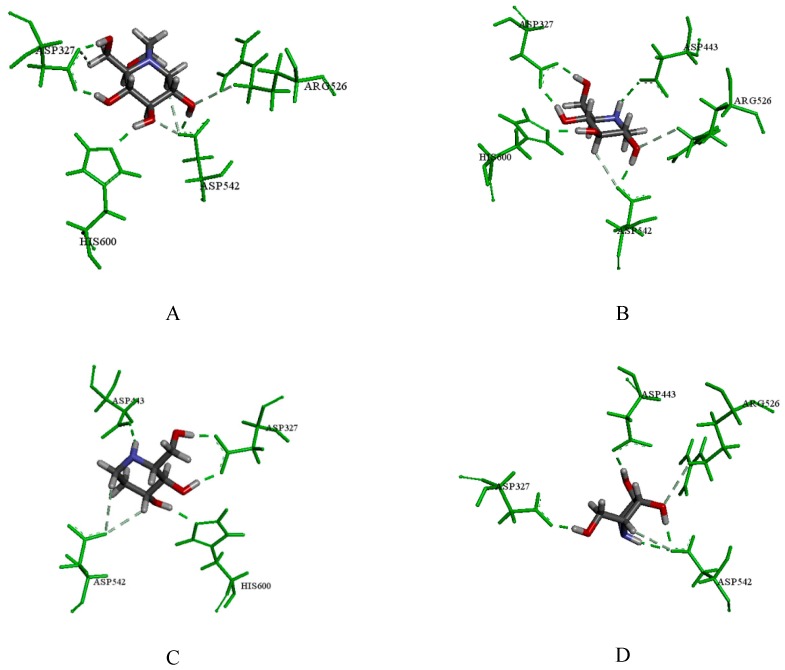
(**A**) The docking result of miglitol with the crystal structure of NtMGAM; (**B**) the docking result of 1-DNJ; (**C**) the docking result of FA; (**D**) the docking result of DAB; the green amino acids represent hydrogen bond interactions.

**Figure 8 molecules-24-01776-f008:**
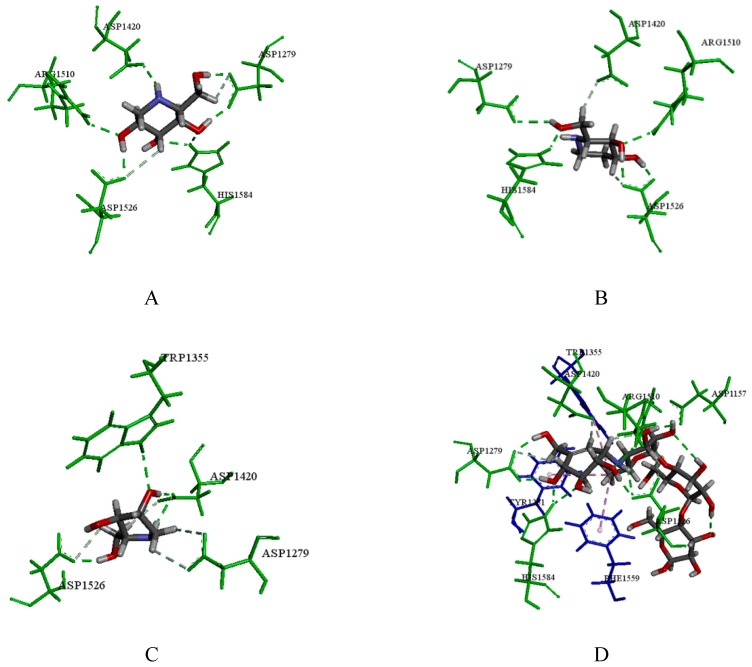
(**A**) The docking result of 1-DNJ with the crystal structure of CtMGAM; (**B**) the docking result of FA; (**C**) the docking result of DAB; (**D**) the docking result of acarbose; the green amino acids represent hydrogen bond interactions; blue amino acids represent hydrophobic interactions; the red amino acids represented unfavorable interactions.

**Table 1 molecules-24-01776-t001:** α-Glucosidase inhibitory activity of compounds.

Samples	Sucrase Inhibition		Maltase Inhibition		α-Amylase Inhibition	
	Inhibition	Ic_50_	Inhibition	IC_50_	Inhibition	IC_50_
	Ratio (%)	(μmol/L)	Ratio (%)	(μmol/L)	Ratio (%)	(μmol/L)
1-DNJ	100	0.078	100	0.113	0	>100
FA	84.6	13.53	65.7	25.97	0	>100
DAB	79.7	21.27	77.9	12.33	0	>100
Acarbose	100	1.046	98.0	0.104	93.6	2.69
SZ-A	100	0.03 μg/mL	100	0.06 μg/mL	0	>100 μg/mL

Note: The inhibitory rate of different compounds on sucrase, maltase and α-amylase was calculated at the concentrations of 4 × 10^−5^ M, 4 × 10^−5^ M and 1 × 10^−5^ M, respectively.

**Table 2 molecules-24-01776-t002:** Kinetic parameters of various inhibitors on maltase and sucrose.

Enzyme	Inhibitor	Inhibition Type	Km (M)	Ki (M)
1-DNJ	maltase	Competitive inhibition	2.14 × 10^−2^	1.12 × 10^−6^
FA	maltase	Competitive inhibition	2.14 × 10^−2^	3.24 × 10^−4^
DAB	maltase	Competitive inhibition	2.14 × 10^−2^	5.21 × 10^−5^
Acarbose	maltase	Competitive inhibition	2.14 × 10^−2^	1.75 × 10^−6^
1−DNJ	sucrose	Competitive inhibition	4.85 × 10^−2^	1.37 × 10^−8^
FA	maltase	Competitive inhibition	4.85 × 10^−2^	3.97 × 10^−6^
DAB	maltase	Competitive inhibition	4.85 × 10^−2^	1.91 × 10^−5^
acarbose	maltase	Competitive inhibition	4.85 × 10^−2^	5.32 × 10^−8^

Note: 1-DNJ, 1-deoxynojirimycin; FA, fagomine; DAB,1,4-dideoxy-1,4-imino-d-arabinitol.

**Table 3 molecules-24-01776-t003:** Solvent accessibility surface of catalytic subunits of different α-glucosidases.

Protein		PDB Code	Ligand	Solvent Accessibility Surface (Å^2^)	Resolution (Å)
HPA	Apo	1HNY	-	17,891.6	1.8
	Complex	3OLD	Acarviostatin I03	17,413.6	2.0
NtSI	Apo	3LPO	-	30,056.4	3.2
	Complex	3LPP	Kotalanol	30,331.4	3.2
NtMGAM	Apo	2QLY	-	30,738	2.0
	Complex	2QMJ	acarbose	29,942.5	1.9
		3L4W	Miglitol	31,044.5	2.0
CtMGAM	Apo	3TON	-	31,142.5	2.95
	Complex	3TOP	acarbose	30,812.4	2.88

Note: NtMAGM, maltase-glucoamylase N-terminal subunit; NtSI, sucrase-isomaltase N-terminal subunit; CtMGAM, maltase-glucoamylase C-terminal subunit; HPA, pancreatic amylase.

**Table 4 molecules-24-01776-t004:** The information of four crystal structures and docking analysis of co-crystalized ligands.

Name	PDB	Ligand	Radius (Å)	RMSD (Å)
HPA	3OLD	Acarviostatin I03	13.70	2.70
NtSI	3LPP	Kotalanol	8.21	1.06
NtMGAM	3L4W	Miglitol	5.66	0.92
CtMGAM	3TOP	acarbose	9.13	1.55

**Table 5 molecules-24-01776-t005:** Evaluation of the interaction between the compounds and the active site of different α-glucosidases.

Name	Consensus (Number of Votes)				Consensus (Normalized Average)			
	HPA	NtMGAM	CtMGAM	NtSI	HPA	NtMGAM	CtMGAM	NtSI
1-DNJ	2	10	11	8	0.27	0.93	0.70	0.89
DAB	3	1	2	5	0.40	0.63	0.54	0.73
Fagomine	2	5	1	3	0.38	0.74	0.55	0.62
acarbose	11	7	10	7	0.93	0.62	0.84	0.75
Miglitol	5	11	10	6	0.35	0.81	0.67	0.76

Note: 1-DNJ, 1-deoxynojirimycin; DAB, 1,4-dideoxy-1,4-imino-d-arabinitol; HPA, pancreatic amylase; NtMGAM, maltase-glucoamylase N-terminal subunit; CtMGAM, maltase-glucoamylase C-terminal subunit; NtSI, sucrase-isomaltase N-terminal subunit

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
