# Peer review of "Investigation on the Enzymatic Profile of Mulberry Alkaloids by Enzymatic Study and Molecular Docking"

_molecules, 2019, doi:10.3390/molecules24091776_

Round 1
Reviewer 1 Report
The study reports the results of the enzymatic profiling of SZ-A a mulberry alkaloids (Sangzhi alkaloids) extracted from the traditional Chinese herbs (Ramulus Mori), proposed as a novel hypoglycemic drug. The clinical results suggested that SZ-A displayed a selective inhibitory effect on disaccharidases, which had great potential for clinical applications and this makes this research interesting.
The study contains also a docking investigation of the major components of SZ-A on the most important enzymes involved in.
In the introduction the composition of the natural extract is outlined. Nevertheless I think that a most detailed information should be given about the minor components.
The AAs defined these (including 1-deoxynojirimycin (1-DNJ), fagomine (FA), 1,4-dideoxy-1,4-imino-darabinitol (DAB), 3-epi-fagomine, 2-o-β-Glc-DAB, 6-o-β-d-Glc-DNJ, and 2-o-β-Glc-DAB as “ effective” but they should define as “ ineffective” the 20% of components of SZ-A. Are they able to do this?.
The reference should be corrected from [13-14] to 13 only.
The study has been conducted in parallel with acarbose as positive control.
Therefore, it is of great significance to research the selective inhibitory activity and mechanism of SZ-A and its major active ingredients (1-DNJ, FA and DAB) in detail.
The α-glucosidase inhibitory profile of SZ-A and its major active
ingredients (1-DNJ, FA and DAB) by enzyme inhibition assay and kinetic studies in vitro. The tresults are interesting but the AAs did not investigated the effects of a combination of the compounds into a mixture similar to that found in SZ-A.
In particular the discussion about possible synergistic effects is absent.
The molecular docking study was performed to provide valuable insights into the binding properties between the major active ingredients (1-DNJ, FA and DAB) and the a-glucosidase and pancreatic amylase.
The AAs should illustrate the criteria applied by the several consensus scoring functions utilized for a less familir reader.
The Figures of the docking are badly labeled. It is hard to read there. The labels and the hydrogen bond are not easily distinguishable.
Moreover in the text aminoacids should be cited as Arg His Asp etc and not ARG1510, HIS1584, ASP1420, ASP1526 and ASP1279.
Also later “ hydrogen bonds with ARG1510, HIS1584, ASP1526, ASP1420 and ASP1279 (Figure 8B).”
And in the following
“DAB formed the hydrogen bonds with TRP1355, ASP1420, ASP1526 and ASP1279 (Figure 8C).
Which means “ Compared with the docking result of acarbose, although an unfavourable interaction with Trp1523 was found, it could form the hydrogen bonds with ARG1510, HIS1584, ASP1526, ASP1157, ASP1420 272 and ASP1279, also formed the hydrophobic interactions with residues TYR1251, TRP1355, and PHE1559, which resulted in a higher affinity for CtMGAM (Figure 8D).”
This sentence should be better detailed . To which hydrophobic interactions the AAs refer to?. Maybe between the electron clouds of aromatic residues?
TheAAs say
“Therefore, our scoring and enzyme inhibition assay results were not uniformly in agreement to some extent. However, on the basis of the above analysis, the
active compounds showed a good correlation between scoring results in the binding mode of disaccharidases (maltase and sucrase) and enzyme inhibition activity in the assays, which further highlighted the importance of molecular mechanic-based scoring function in revealing the biological properties of target compounds.”
This sentence should be more clear and detailed. These results are in agreement or not?.The results of clinical trials cannot be used to enforce the investigation at molecular level.
About the several Consensus scoring reported in M&M the comment has been reported above.
Minor point .
There are several text errors. Some word are in blue and underlined from row 145 to 147 and from 212 to 220 ( why?).
The AAs should decide if to use alpha, a- or a in the citing the enzymes.
At row 77 The ref 13,14 should be corrected in 13 alone.
Author Response
Dear Reviewer 1:
Thank you for your comments concerning our manuscript entitled “Investigation on the enzymatic profile of mulberry alkaloids by enzymatic study and molecular docking”. Those comments are all valuable and very helpful for revising and improving our paper. We have studied comments carefully and have made correction which we hope meet with approval. Revised portion are marked in red in the paper. The main corrections in the paper and the responds to the comments are as following. We would much appreciate it if you could accept our effort and revision. We would be glad to respond to any further questions and comments that you may have.
1. Revision suggestion: The AAs defined these (including 1-deoxynojirimycin (1-DNJ), fagomine (FA), 1,4-dideoxy-1,4-imino-darabinitol (DAB), 3-epi-fagomine, 2-o-β-Glc-DAB, 6-o-β-d-Glc-DNJ, and 2-o-β-Glc-DAB as “ effective” but they should define as “ ineffective” the 20% of components of SZ-A. Are they able to do this?
Response: Thank you very much for your comment. We are sorry that this part was not clear in the original manuscript. In fact, the content of 1-DNJ, FA and DAB accounts for more than 80% of the total alkaloids instead of “SZ–A extract”. We have re-written these sentences in Introduction (Page 2, Line 75-80).
2. Revision suggestion: The reference should be corrected from [13-14] to 13 only.
Response: As your suggestion, the reference has been corrected to [13] only (Page 2, Line 78).
3. Revision suggestion: The α-glucosidase inhibitory profile of SZ-A and its major active ingredients (1-DNJ, FA and DAB) by enzyme inhibition assay and kinetic studies in vitro. The results are interesting but the AAs did not investigated the effects of a combination of the compounds into a mixture similar to that found in SZ-A. In particular, the discussion about possible synergistic effects is absent.
Response: Special thanks to you for your good comment. It is helpful to illustrate the inhibition profile of different compounds. In our prophase research, we found that the total alkaloid content in SZ-A extract is about 55%, the major active ingredients in SZ-A are 1-DNJ, FA and DAB, which account for more than 80% of the total alkaloids, and 1-DNJ account for more than 70%. Compared with FA and DAB, 1-DNJ has the highest content and the strongest inhibition activity on α-glycosidase. It indicated that 1-DNJ was the main contributor to inhibit α-glycosidase, and FA and DAB might only play an auxiliary role. In addition, SZ-A also contains many other components (except alkaloids ) such as polysaccharides, amino acids and flavonoids, which may influence the inhibitory activities of SZ-A against α-glucosidase. Thus, it is very difficult and complex to explore possible synergistic effects and mechanism of different compounds in a short time. Considering the timeliness of our research work, it is not suitable to publish whole study results in single paper. As your suggestion, we will carry out this experiment in the near future and look forward to your attention. And the discussion has been added into Conclusions (Page 13).
4. Revision suggestion: The AAs should illustrate the criteria applied by the several consensus scoring functions utilized for a less familiar reader.
Response: Thank you very much for your reminder. We have carried out a more detailed description of the criteria applied by the several consensus scoring functions in Introduction (Page 3, Line 106-110).
5. Revision suggestion: The Figures of the docking are badly labeled. It is hard to read there. The labels and the hydrogen bond are not easily distinguishable.
Response: We are very sorry for our negligence of this problem. As your suggestion, we have changed more clear figures for Figure 5-8 and the resolution of labels and the hydrogen bond were all improved.
6. Revision suggestion: Moreover in the text aminoacids should be cited as Arg His Asp etc and not ARG1510, HIS1584, ASP1420, ASP1526 and ASP1279. Also later “ hydrogen bonds with ARG1510, HIS1584, ASP1526, ASP1420 and ASP1279 (Figure 8B).”
Response: Thank you for your reminder. As your suggestion, we have corrected the format of all the amino acids in the manuscript.
7. Revision suggestion: “DAB formed the hydrogen bonds with TRP1355, ASP1420, ASP1526 and ASP1279 (Figure 8C). Which means “ Compared with the docking result of acarbose, although an unfavourable interaction with Trp1523 was found, it could form the hydrogen bonds with ARG1510, HIS1584, ASP1526, ASP1157, ASP1420 272 and ASP1279, also formed the hydrophobic interactions with residues TYR1251, TRP1355, and PHE1559, which resulted in a higher affinity for CtMGAM (Figure 8D).” This sentence should be better detailed. To which hydrophobic interactions the AAs refer to?. Maybe between the electron clouds of aromatic residues?
Response: We are sorry for our inappropriate expression. As your suggestion, we have corrected this sentence in the paper. Detailed description was added into Results section 2.3 (Page 10, Line 275-278).
8. Revision suggestion: TheAAs say “Therefore, our scoring and enzyme inhibition assay results were not uniformly in agreement to some extent. However, on the basis of the above analysis, the active compounds showed a good correlation between scoring results in the binding mode of disaccharidases (maltase and sucrase) and enzyme inhibition activity in the assays, which further highlighted the importance of molecular mechanic-based scoring function in revealing the biological properties of target compounds.” This sentence should be more clear and detailed. These results are in agreement or not?. The results of clinical trials cannot be used to enforce the investigation at molecular level.
Response: We are very sorry for our in-corrected writing, the scoring results in the binding mode are in agreement with inhibition activity. We have re-written these sentences in Results section 2.3 (Page 11, Line 292-294). In addition, we accept Reviewer’s comment and delete the description of clinical trials in Conclusions.
9. Revision suggestion: There are several text errors. Some word are in blue and underlined from row 145 to 147 and from 212 to 220 ( why?).
Response: We are very sorry for our negligence of text format, we have made correction according the Reviewer’s comment in Results section 2.2 (Page 4, Line 152) and Results section 2.3 (Page 7, Line219, 224).
10. Revision suggestion: The AAs should decide if to use alpha, a- or a in the citing the enzymes.
Response: As your suggestion, we have corrected the format in Abstract (Page 1, Line 19)
11. Revision suggestion: At row 77 The ref 13,14 should be corrected in 13 alone.
Response: The reference has been corrected to [13] alone.
Reviewer 2 Report
The authors characterized the mulberry alkaloids by steady-state kinetic analysis and molecular docking. The experimental protocol was well-designed and well-conducted. I think that the manuscript is acceptable if the authors respond to the following minor points.
Minor points:
1. Is the Ki value of acarbose toward sucrose correct? In Fig. 2, The concentrations of acarbose are 0, 6.3, 12.5, and 25.0 × 10-6 M. It is difficult to obtain Ki value of 5.32× 10-8 M from this experiment.
2. In Table 2, the Km value indicates the one without inhibitor. The current table lays readers open to misleading.
3. In line 28, “sucrose” should be “sucrase”. In line 53, “maltose, sucrose” should be “maltase, sucrase”.
Author Response
Dear Reviewer 2:
Thank you for your comments concerning our manuscript entitled “Investigation on the enzymatic profile of mulberry alkaloids by enzymatic study and molecular docking”. Those comments are all valuable and very helpful for revising and improving our paper. We have studied comments carefully and have made correction which we hope meet with approval. Revised portion are marked in red in the paper. The main corrections in the paper and the responds to the comments are as following. We would much appreciate it if you could accept our effort and revision. We would be glad to respond to any further questions and comments that you may have.
1. Revision suggestion: Is the Ki value of acarbose toward sucrose correct? In Fig. 2, The concentrations of acarbose are 0, 6.3, 12.5, and 25.0 × 10-6 M. It is difficult to obtain Ki value of 5.32× 10-8 M from this experiment.
Response: We are very sorry for our negligence of this problem. The concentrations of acarbose are 0, 6.3, 12.5, and 25.0 × 10-8 M, we have corrected this problem in Results section 2.2 (Page 5, Line 166-167).
2. Revision suggestion: In Table 2, the Km value indicates the one without inhibitor. The current table lays readers open to misleading.
Response:. Thank you for your reminder. As your suggestion, we have corrected the table lays in
Results section 2.2 ( Table 2)
3. Revision suggestion: In line 28, “sucrose” should be “sucrase”. In line 53, “maltose, sucrose” should be “maltase, sucrase”.
Response: We are very sorry for our incorrected writing. As your suggestion, we have corrected these problems in Abstract (Page 1, Line 28) and Introduction (Page 2, Line 54, 55)